# Synthesis and Characterization of Metal Modified Catalysts for Decomposition of Ibuprofen from Aqueous Solutions

**Soudabeh Saeid** [1] , **Matilda Kråkström** [2], **Pasi Tolvanen** [1] , **Narendra Kumar** [1,*], **Kari Eränen** [1], **Markus Peurla** [3], **Jyri-Pekka Mikkola** [1,4], **Laurent Maël** [1], **Leif Kronberg** [2], **Patrik Eklund** [2] **and Tapio Salmi** [1,*]

[1] Laboratory of Industrial Chemistry and Reaction Engineering, Johan Gadolin Process Chemistry Centre, Åbo Akademi University, Biskopsgatan 8, FI-20500 Åbo/Turku, Finland; soudabeh.saeid@abo.fi (S.S.); pasi.tolvanen@abo.fi (P.T.); kari.eranen@abo.fi (K.E.); jyri-pekka.mikkola@abo.fi (J.-P.M.); mael.laurent@insa-rouen.fr (L.M.)

[2] Laboratory of Organic Chemistry, Johan Gadolin Process Chemistry Centre, Åbo Akademi University, Biskopsgatan 8, FI-20500 Åbo/Turku, Finland; matilda.kråkström@abo.fi (M.K.); leif.kronberg@abo.fi (L.K.); patrik.j.eklund@abo.fi (P.E.)

[3] Laboratory of Electron Microscopy, Institute of Biomedicin, University of Turku, FI-20540 Turku, Finland; markus.peurla@utu.fi

[4] Technical Chemistry Department of Chemistry Chemical-Biological Center, Umeå University, SE-90187 Umeå, Sweden

\* Correspondence: nkumar@abo.fi (N.K.); tsalmi@abo.fi (T.S.); Tel.: +358-443-458-107 (N.K.); +358-221-544-27 (T.S.)

**Abstract:** The presence of pharmaceuticals in surface water, drinking water, and wastewater has attracted significant concern because of the non-biodegradability, resistance, and toxicity of pharmaceutical compounds. The catalytic ozonation of an anti-inflammatory pharmaceutical, ibuprofen was investigated in this work. The reaction mixture was analyzed and measured by high-performance liquid chromatography (HPLC). Liquid chromatography-mass spectrometry (LC-MS) was used for the quantification of by-products during the catalytic ozonation process. Ibuprofen was degraded by ozonation under optimized conditions within 1 h. However, some intermediate oxidation products were detected during the ibuprofen ozonation process that were more resistant than the parent compound. To optimize the process, nine heterogeneous catalysts were synthesized using different preparation methods and used with ozone to degrade the ibuprofen dissolved in aqueous solution. The aim of using several catalysts was to reveal the effect of various catalyst preparation methods on the degradation of ibuprofen as well as the formation and elimination of by-products. Furthermore, the goal was to reveal the influence of various support structures and different metals such as Pd-, Fe-, Ni-, metal particle size, and metal dispersion in ozone degradation. Most of the catalysts improved the elimination kinetics of the by-products. Among these catalysts, Cu-H-Beta-150-DP synthesized by the deposition–precipitation process showed the highest decomposition rate. The regenerated Cu-H-Beta-150-DP catalyst preserved the catalytic activity to that of the fresh catalyst. The catalyst characterization methods applied in this work included nitrogen adsorption–desorption, scanning electron microscopy, transmission electron microscopy, and Fourier-transform infrared spectroscopy. The large pore volume and small metal particle size contributed to the improved catalytic activity.

**Keywords:** advanced oxidation process; zeolites; catalyst preparation; catalyst characterization; wastewater treatment

## 1. Introduction

Due to a globally increasing consumption of pharmaceuticals in the recent years, a pharmaceutical cocktail has emerged in surface waters and effluents from human communities. The detection of pharmaceutical species has caused a significant concern among scientists and laymen associations because the slip of pharmaceuticals has a negative effect on the environment [1,2]. In the current decade, various studies have been published concerning the appearance and ecological hazard of pharmaceuticals and personal care compounds released to the environment. Some of these pharmaceuticals pose a high risk to the marine life and humankind [3–5]. Pharmaceuticals can have a serious effect even at very low concentration levels, because they are designed to have a high biological activity at low doses to perform specific mechanisms in humans and animals [6]. These particles can mainly enter aquatic systems directly from pharmaceutical factories, hospitals, and households as well as livestock [7,8]. Pharmaceuticals are hardly decomposed at all by conventional water treatment due to their chemical stability. Among these treatments, biodegradation is one of the valid methods for removing pharmaceuticals. However, for pharmaceuticals that are resistant to biodegradation processes, an advanced oxidation process (AOP) is necessary [9].

AOPs provide an excellent potential for the destructive treatment of organic compounds such as pharmaceutical residues. These processes imply the mineralization of organic components to $CO_2$ via highly reactive and nonselective species, i.e., hydroxyl radicals (HO·), $H_2O_2$, and $O_3$ [10]. Several AOP processes for this purpose are for instance, ozone-based processes [11], Fenton and photo-Fenton [12], UV, UV/$H_2O_2$ [13], as well as electrochemical oxidation [14]. For producing potable water, an addition of disinfection chemicals is needed. Some of the conventional disinfectants are chlorine, chloramines, and ozone [15]. It is known that chlorine applied for the disinfection of drinking water is able to react with organic contaminates present in water and generate by-products such as chloroform. This has directed to employ alternative disinfection processes such as catalytic ozonation [16].

Ozone is generally employed in water treatment due to its solubility, reactivity as well as electrophilic and nucleophilic characteristics. Ozone is a strong oxidant; however, it has some limitations, because it reacts with some organic and inorganic compounds (e.g., saturated aliphatic acid and $NH_4^+$). To reach a high degree of mineralization, AOPs including ozone could be utilized, for example $O_3$/$H_2O_2$, $O_3$/UV, $O_3$/UV/$H_2O_2$ and catalytic ozonation, which can create more effective radicals such as HO· [17]. The advantage with the ozonation process is that it can be applied at ambient pressure and temperature; furthermore, this process creates unselective hydroxyl radicals, which are able to eliminate micropollutants such as personal care products and pharmaceuticals from waste waters. Among the ozone-based technologies, one of the most recommended approaches to improve the purification performance and to achieve a higher level of mineralization is the combination of ozone and an efficient and durable catalyst [18].

Catalytic ozonation can be feasible technology for the elimination of an extensive range of contaminants from industrial wastewaters and pharmaceuticals in wastewaters. Nevertheless, catalytic ozonation is mostly utilized at the laboratory scale. This kind of treatment has successful outcomes and it consists of homogeneous and heterogeneous catalytic ozonation. In homogeneous catalytic ozonation, ozone can be activated through metal ions existing in water. On the other hand, in heterogeneous catalytic ozonation, ozone can be activated via metal oxides upon supports. Typical solid materials used in heterogeneously catalyzed ozonation are zeolites, metal oxides supported on zeolites, and carbon compounds [19,20].

Copper has been applied as a catalytic material during the last few decades due to its redox character, recyclability, and low price in a number of industrial processes. Various methods have been utilized for the synthetization of copper-based catalysts [21–23]. For example, Xin et al. studied the CuO/SBA-15 catalyst preparation through the deposition–precipitation process. This method displayed an effective and scalable way of fabricating a copper-based catalyst with a desirable oxidation activity [21].

Ibuprofen [2-(4-isobutyl phenyl) propionic acid] (IBU) is generally prescribed for suppressing inflammation, pain, and fever [24,25]. IBU is prepared in several formulations and manufactured in high volumes. Furthermore, IBU is one of the primary pharmaceuticals placed in the list of essential drugs of the World Health Organization (WHO) [26]. IBU is one of the over-the-counter used painkillers, and it is frequently combined with other conventional medicines, containing antihistamines and decongestants [27]. Residues of IBU have been widely detected in surface and ground waters [28]. For example, IBU was found at high concentrations in the effluent and receiving water of waste water treatment plants (WWTPs) in the River Aire and Calder catchments in the UK, and the maximum concentration of detected IBU was 4.83 µg/L [29]. Lindqvist et al. described that IBU is the most frequently used pharmaceutical compound in Finland, and it is one of the highest detected compounds in the raw sewage. This compound was found at the discharge points of the sewage treatment effluents in rivers, because of the meager removal of the sewage treatment plants. In their research, concentrations of 13.1 µg/L IBU were discovered in the influent of sewage treatment [30]. The occurrence of IBU in aquatic ecosystems has been associated with several toxic impacts on marine organisms. For the fresh water fish Rhamdia quelen, an IBU exhibition for a long duration (14 days) but at low concentration can induce health effects [26]. The degradation of IBU has been investigated in several studies. For instance, Jin et al. investigated the degradation of IBU by $Fe^{II}$-NTA complex-activated persulfate including hydroxylamine, which demonstrated the successful degradation of IBU. However, hydroxylamine is a toxic agent, and it is not environmentally friendly [31]. Moreover, Xiang et al. investigated the IBU degradation applying the combination of UV and chlorine, which showed a high first-order rate constant. Nevertheless, the controversial issue of this method was the toxicity of the chlorinated by-products [32]. The abiotic degradation of IBU and toxic impacts of basic ibuprofen and its secondary residues reveals the various grade of toxicity of these pharmaceuticals [33]. Accordingly, it is very urgent to discover a practically applicable method for removing IBU without or with a small amount of disinfectants or hazardous by-products. The elimination of IBU by catalytic ozonation using multi-walled carbon nanotubes was investigated by Du et al., who revealed that this catalyst improved the removal of IBU because the catalyst enhanced the $HO^{\bullet}$ formation [34]. This study confirmed that catalytic ozonation could be a beneficial method for the elimination of IBU; however, unfortunately, nothing was reported about the formation and transformation of the by-products of this reaction.

In the previous work of our group, the degradation of IBU with either non-catalytic, or with H- and Fe-modified Beta zeolite catalyzed ozonation has been studied. Besides optimizing the degradation process and to increase the ozone concentration in water, different experimental parameters were examined. However, in the previous study, IBU was degraded entirely in three hours of catalytic ozonation under optimal conditions, but by-products were not studied [35]. Based on previous experience, the present work was designed to achieve the total decomposition of IBU and transformation products of IBU in a shorter time of ozonation. For this purpose, different nitrogen concentrations were used in the inlet gas of ozonator, and nine different catalysts (Cu-H-Beta-25-IE, Cu-H-Beta-150-IE, Cu-H-Beta-300, Cu-H-Beta-150-EIM, Cu-H-Beta-150-DP, Cu-Na-Mordenite-12.8-IE, Pd-H-MCM-41-EIM, Fe-SiO$_2$-DP, and Ni-H-Beta-25-EIM) were used for the removal of IBU. The intermediates and by-products formed in these experiments were studied and tentatively identified by liquid chromatography-mass spectrometry (LC-MS/MS). In addition, the catalysts were characterized by several methods.

## 2. Results and Discussion

### 2.1. Physico-Chemical Characterization

#### 2.1.1. Transmission Electron Microscopy (TEM)

To study the particle size distributions and structures of Cu, Pd, Fe, and Ni-based catalysts, high-resolution transmission electron microscopy (TEM) was used. TEM micrographs of Cu-H-Beta-25-IE, Cu-H-Beta-150-IE, Cu-H-Beta-300-IE, Cu-H-Beta-150-EIM, Cu-H-Beta-150-DP,

Cu-Na-Modernite-12.8-IE, Pd-H-MCM-41-EIM, Fe-SiO$_2$-DP, and Ni-H-Beta-25-EIM as well as the Cu, Pd, Fe, and Ni particle size distributions, given as histograms, are displayed in Figure 1a–i.

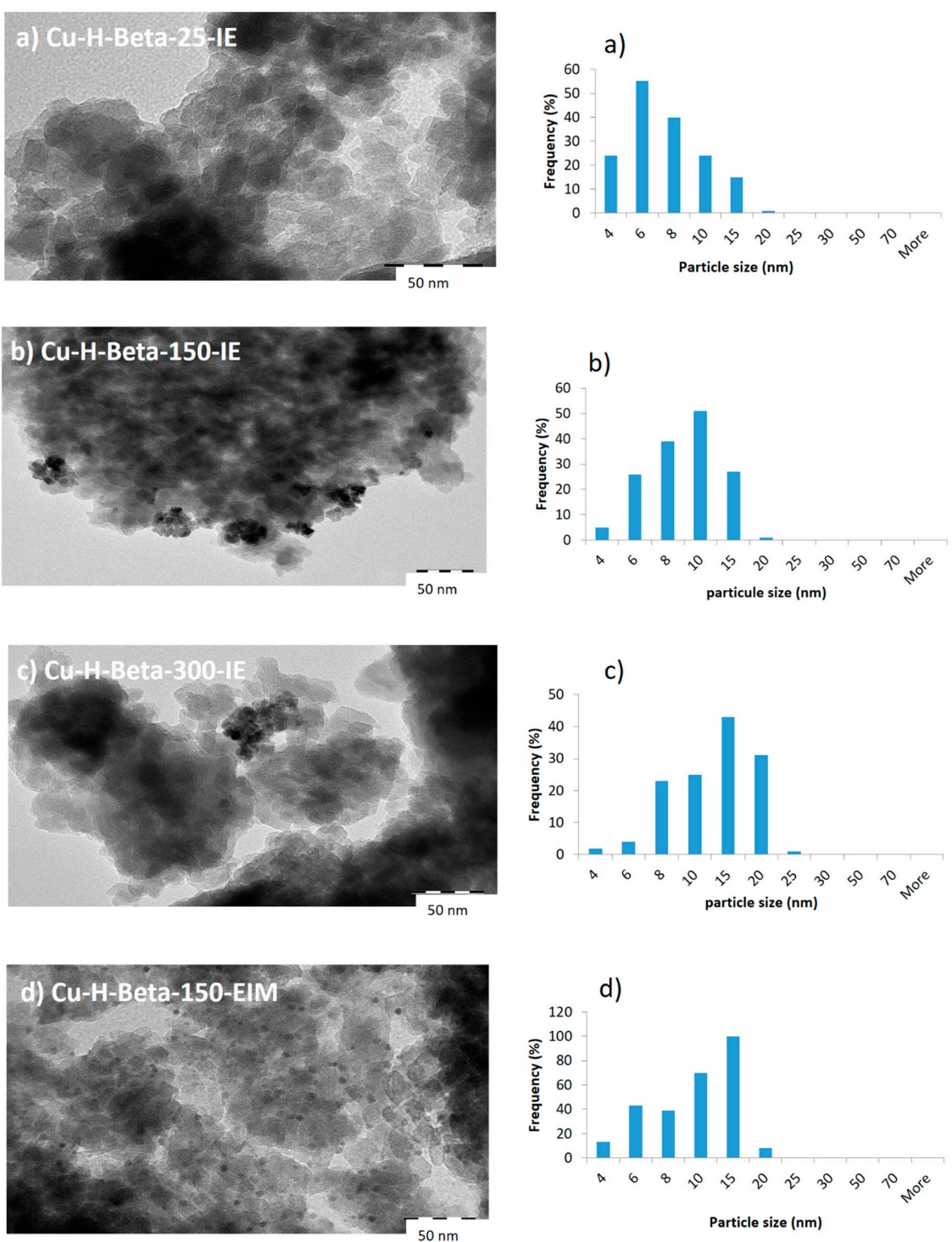

**Figure 1.** *Cont.*

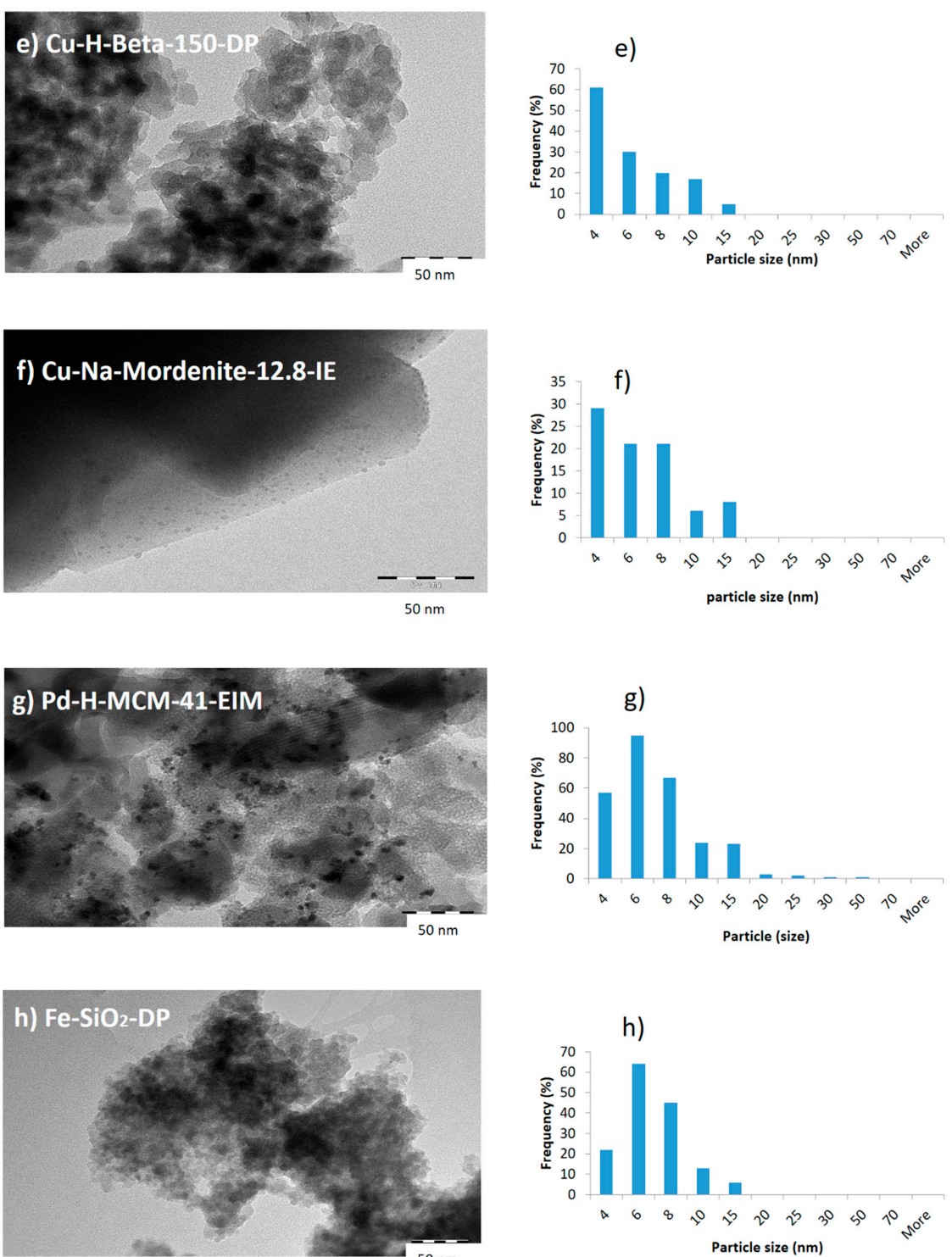

**Figure 1.** *Cont.*

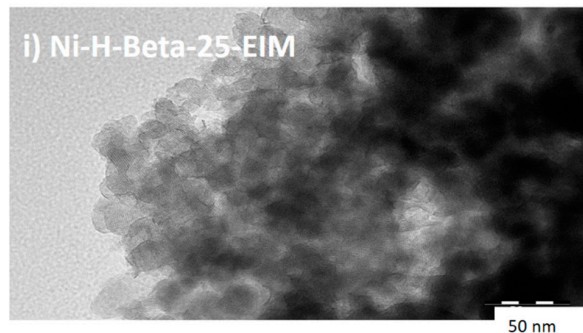 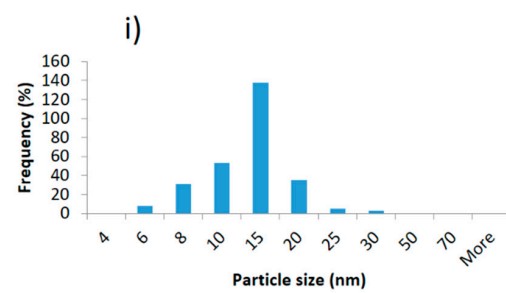

**Figure 1.** Transmission electron microscope (TEM) images and Cu, Pd, Fe, Sn, Ni particle size distribution histograms of Cu-H-Beta-25-IE, Cu-H-Beta-150-IE, Cu-H-Beta-300-IE, Cu-H-Beta-150-EIM, Cu-H-Beta-150-DP, Cu-Na-Mordenite-12.8-IE, Pd-H-MCM-41-EIM, Fe-SiO$_2$-DP, and Ni-H-Beta-25-EIM (**a**–**i**).

The Cu, Pd, Fe, and Ni average particle size and particle size distributions were determined by TEM (Table 1). The largest average Cu particle size (11.49 nm) was determined for Cu-H-Beta-300-IE. It was observed that the method of the catalyst synthesis influenced the average Cu crystal size. Thus, Cu-H-Beta-150-DP illustrated the smallest average Cu particle size (4.88 nm). The average particle size of Pd, Fe, and Ni were measured to be 6.31 nm, 5.86 nm, and 11.61 nm, respectively.

**Table 1.** Cu, Pd, Fe Sn, Ni particle size average of Cu-H-Beta-25-IE, Cu-H-Beta-150-IE, Cu-H-Beta-300-IE, Cu-H-Beta-150-EIM, Cu-H-Beta-150-DP, Cu-Na-Mordenite-12.8-IE, Pd-H-MCM-41-EIM, Fe-SiO$_2$-DP, and Ni-H-Beta-25-EIM catalysts.

| Entry | Catalyst | Average Metal Particle Size (nm) |
|:---:|:---:|:---:|
| 1 | Cu-H-Beta-25-IE | 6.42 |
| 2 | Cu-H-Beta-150-IE | 8.13 |
| 3 | Cu-H-Beta-300-IE | 11.49 |
| 4 | Cu-H-Beta-150-EIM | 9.22 |
| 5 | Cu-H-Beta-150-DP | 4.88 |
| 6 | Cu-Na-Mordenite-12.8-IE | 5.56 |
| 7 | Pd-H-MCM-41-EIM | 6.31 |
| 8 | Fe-SiO$_2$-DP | 5.86 |
| 9 | Ni-H-Beta-25-EIM | 11.61 |

### 2.1.2. Nitrogen Physisorption

The specific surface areas and pore volumes of the catalysts were analyzed by nitrogen adsorption–desorption (Table 2). The specific surface areas and pore volumes of fresh, spent, and regenerate catalysts are presented in Table 3. The lowest specific surface area was determined for Fe-SiO$_2$-DP (305 m$^2$/g) and the highest was presented for Cu-H-Beta-300-IE (1013 m$^2$/g) catalyst. The Cu-H-Beta-150-EIM spent catalysts showed a decrease in the surface area (470 m$^2$/g) as compared to Cu-H-Beta-150-EIM-Fresh (846 m$^2$/g) catalyst (Table 3). This evidence might be due to the adsorption of produced organic intermediates or the oxidation of the catalyst surface via ozone [36,37]. However, the Cu-H-Beta-150-EIM and Cu-H-Beta-DP spent catalysts were successfully regenerated, the surface areas of the regenerated Cu-H-Beta-150-EIM and Cu-H-Beta-150-DP were determined to 537 m$^2$/g and 640 m$^2$/g, respectively, which confirms that a good recovery was obtained for Cu-H-Beta-150-DP. The surface areas of the regenerated catalysts were not the same as those for the fresh catalysts, and they were slightly decreased possibly because of the thermal regeneration of the ozone-oxidized surfaces. In earlier research on the catalytic ozonation of pharmaceutical compound diclofenac, it was observed that the amount of metal loading was very important and effective on the MCM-41 catalyst activity. For this reason, the metal loading was examined by us. The catalyst activity was enhanced via metal loading; however, overloading the metal can block the pores and active sites of the catalysts, resulting

in a dramatic decline of the catalytic activity. Consequently, the metal content was kept small for most of the synthesized catalysts [38].

**Table 2.** Specific surface area, pore volume, and metal (Cu, Pd, Fe, Ni) content of the catalysts employed in the ozonation experiments.

| Entry | Catalyst | Specific Surface Area ($m^2$/g) | Pore Specific Volume ($cm^3$/g) | Metal Concentration (wt %) |
|---|---|---|---|---|
| 1 | Cu-H-Beta-25-IE | 694 | 0.246 | 1.13 |
| 2 | Cu-H-Beta-150-IE | 542 | 0.192 | 1.34 |
| 3 | Cu-H-Beta-300-IE | 1013 | 0.359 | 0.53 |
| 4 | Cu-H-Beta-150-EIM | 846 | 0.300 | 7.34 |
| 5 | Cu-H-Beta-150-DP | 731 | 0.259 | 6.19 |
| 6 | Cu-Na-Mordenite-12.8-IE | 446 | 0.158 | 3.93 |
| 7 | Pd-H-MCM-41-EIM | 699 | 0.411 | 2.23 |
| 8 | Fe-SiO$_2$-DP | 305 | 0.504 | 6.52 |
| 9 | Ni-H-Beta-25-EIM | 567 | 0.201 | 10.31 |

**Table 3.** Specific surface area, pore volume of the fresh, spent, and regenerated Cu-H-Beta-150-EIM, Cu-H-Beta-150-DP catalysts.

| Entry | Catalyst | Specific Surface Area ($m^2 \cdot g^{-1}$) | | | Pore Specific Volume ($cm^3 \cdot g^{-1}$) | | |
|---|---|---|---|---|---|---|---|
| | | Fresh | Spent | Regenerated | Fresh | Spent | Regenerated |
| 4 | Cu-H-Beta-150-EIM | 846 | 470 | 537 | 0.300 | 0.1672 | 0.1909 |
| 5 | Cu-H-Beta-150-DP | 731 | 548 | 640 | 0.259 | 0.1948 | 0.227 |

### 2.1.3. Energy Dispersive X-ray Microanalyses (EDXA)

The Cu, Ni, Pd, and Fe metal contents in the metal-modified catalysts were analyzed using energy-dispersive X-ray microanalyses (EDXA); the analysis was performed three times for each catalyst, and the average amount was calculated and presented in Table 2. The largest amount of Cu was obtained for the Cu-H-Beta-150-EIM catalyst, the lowest amount of Cu- was obtained for the Cu-H-Beta-25-IE catalyst, and the largest metal content was obtained for Ni-H-Beta-25-EIM (Table 2).

### 2.1.4. Scanning Electron Microscopy (SEM)

The morphologies of the catalysts were studied with scanning electron microscopy (SEM) using a Zeiss Leo Gemini 1530 microscope. SEM reveals the crystal size, shape, and distribution. The crystal size distribution of the (a) Cu-H-Beta-25-IE, (b) Cu-H-Beta-150-IE, (c) Cu-H-Beta-300-IE, (d) Cu-H-Beta-150-EIM, (e) Cu-H-Beta-150-DP, (f) Cu-Na-Mordenite-12.8-IE, (g) Pd-H-MCM-41-EIM, (h) Fe-SiO$_2$-DP, and (i) Ni-H-Beta-25-EIM are presented in Figure 2a–i.

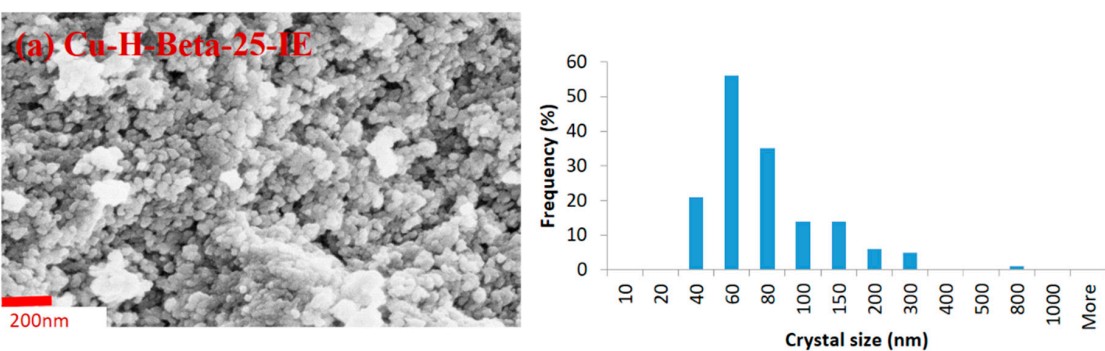

**Figure 2.** *Cont.*

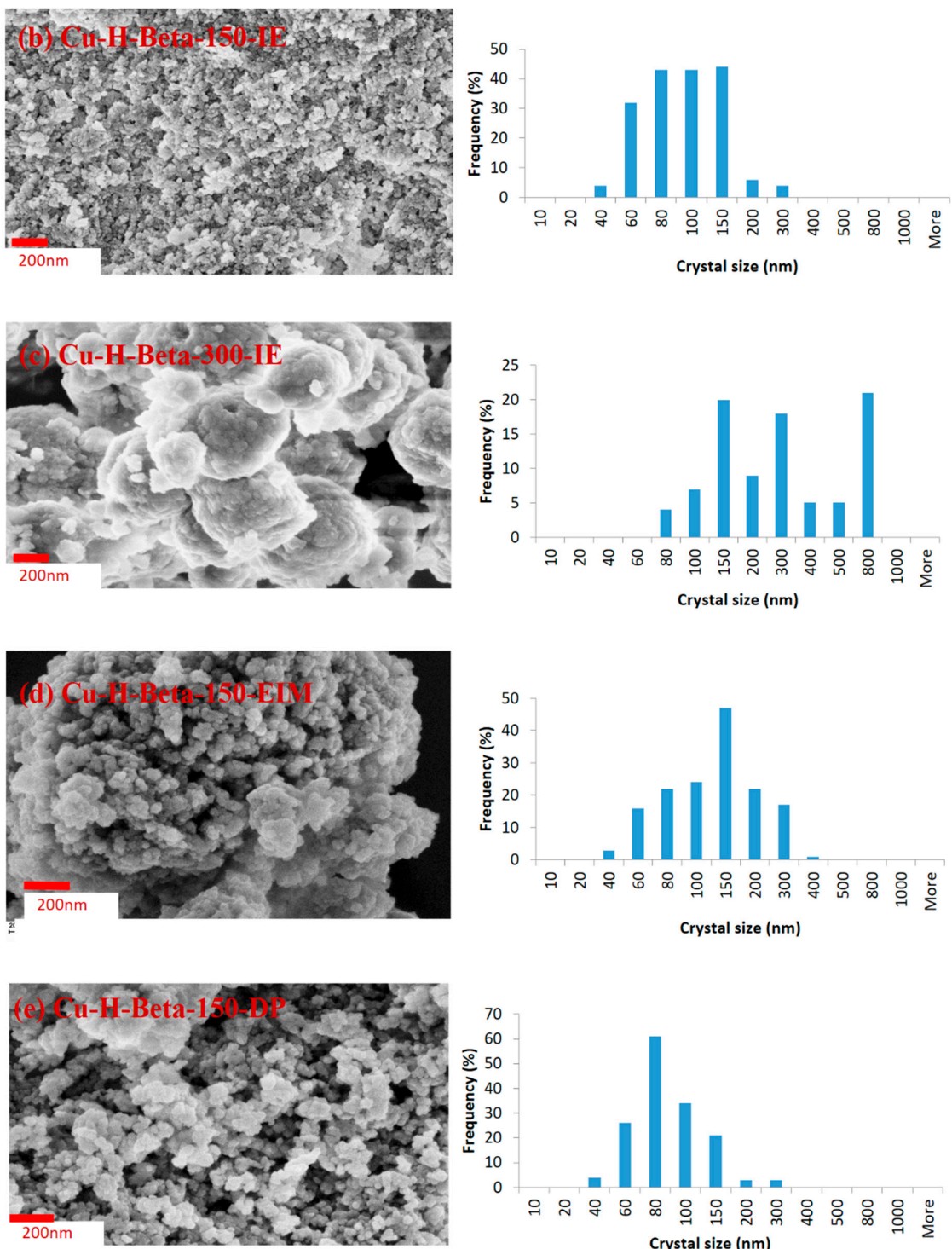

**Figure 2.** *Cont.*

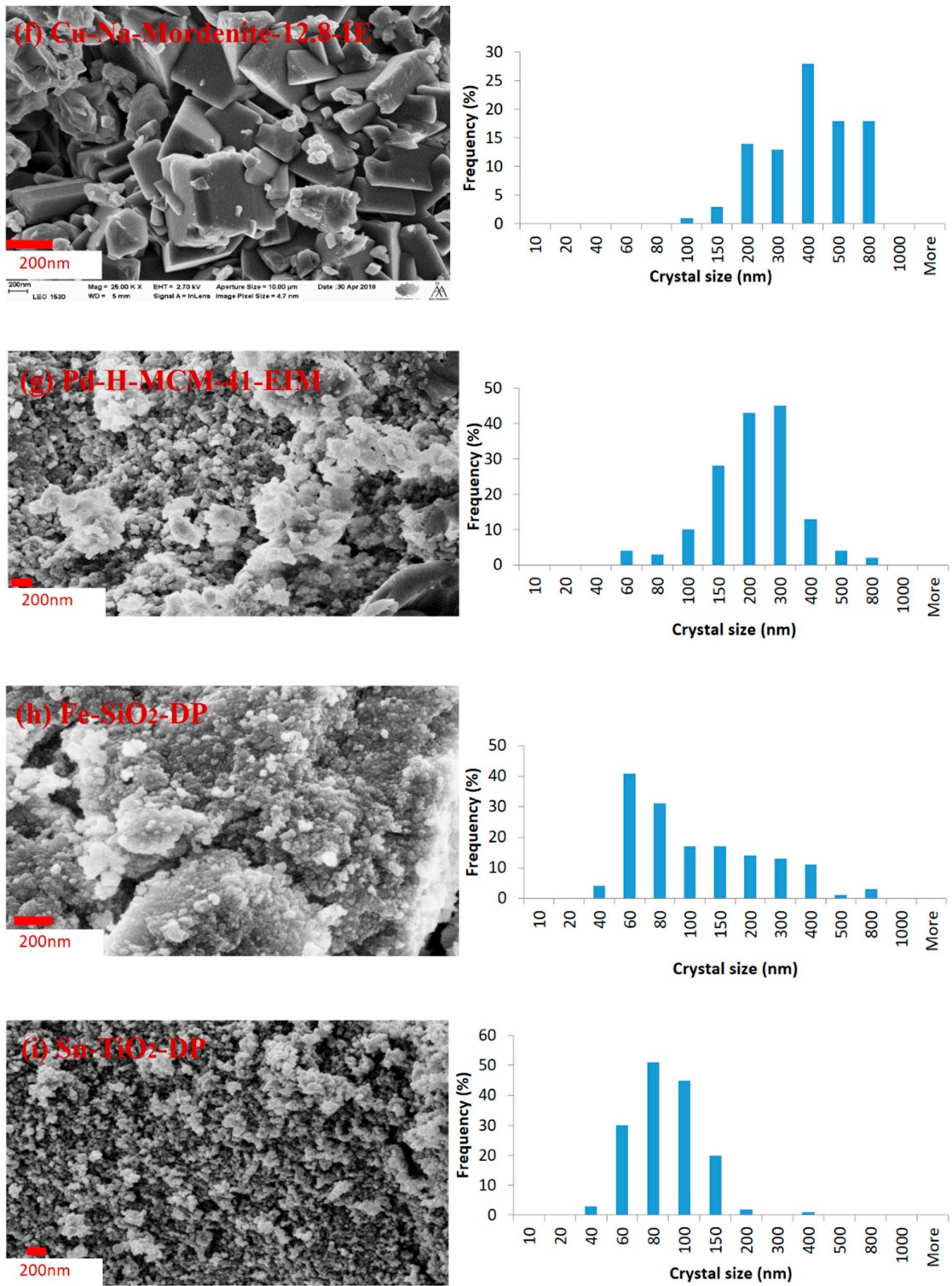

**Figure 2.** Scanning electron micrographs and crystal size distribution histograms of (**a**) Cu-H-Beta-25-IE, (**b**) Cu-H-Beta-150-IE, (**c**) Cu-H-Beta-300-IE, (**d**) Cu-H-Beta-150-EIM, (**e**) Cu-H-Beta-150-DP, (**f**) Cu-Na-Modernite-12.8-IE, (**g**) Pd-H-MCM-41-EIM, (**h**) Fe-SiO$_2$-DP, and (**i**) Ni-H-Beta-25-EIM catalysts.

The crystal sizes of the catalysts were measured, and the size distributions are given in the form of histograms. The average sizes of the crystals of all the studied catalysts were calculated, and they are listed Table 4. The largest average crystal size (369.51 nm) was measured for Cu-Na-Mordenite-12.8-IE

(Table 4), while the second largest crystal size (290.42 nm) was measured for Cu-H-Beta-300-IE. The smallest crystal size (76.08 nm) was measured for Cu-H-Beta-25-IE.

**Table 4.** Average crystal size of Cu-H-Beta-25-IE, Cu-H-Beta-150-IE, Cu-H-Beta-300-IE, Cu-H-Beta-150-EIM, Cu-H-Beta-150-DP, Cu-Na-Modernite-12.8-IE, Pd-H-MCM-41-EIM, Fe-SiO$_2$-DP, and Ni-H-Beta-25-EIM catalysts.

| Entry | Catalyst | Average Crystal Size (nm) |
|:---:|:---:|:---:|
| 1 | Cu-H-Beta-25-IE | 76.08 |
| 2 | Cu-H-Beta-150-IE | 88.142 |
| 3 | Cu-H-Beta-300-IE | 290.42 |
| 4 | Cu-H-Beta-150-EIM | 126.05 |
| 5 | Cu-H-Beta-150-DP | 80.87 |
| 6 | Cu-Na-Mordenite-12.8-IE | 369.51 |
| 7 | Pd-H-MCM-41-EIM | 155.70 |
| 8 | Fe-SiO$_2$-DP | 134.59 |
| 9 | Ni-H-Beta-25-EIM | 97.07 |

### 2.1.5. Pyridine Adsorption–Desorption with FTIR Spectroscopy

Brønsted and Lewis acid sites of the proton form and Cu- modified zeolites were analyzed with Fourier transform infrared spectroscopy (FTIR). The amount of the Brønsted and Lewis acid sites of the proton form and Cu-H-Beta-25-IE, Cu-H-Beta-150-IE, and Cu-H-Beta-300-IE catalysts are presented in Table 5 [39].

**Table 5.** Brønsted and Lewis acidities of the proton and Cu modified Beta zeolites [39].

| Catalysts | Brønsted Acidity (µmol/g) | | | Lewis Acidity (µmol/g) | | |
|:---:|:---:|:---:|:---:|:---:|:---:|:---:|
| | 250 °C | 350 °C | 450 °C | 250 °C | 350 °C | 450 °C |
| **H-Beta-25** | 269 | 207 | 120 | 162 | 128 | 113 |
| **Cu-H-Beta-25-IE** | 136 | 211 | 67 | 180 | 35 | 3 |
| **Cu-H-Beta-150-IE** | 153 | 170 | 113 | 179 | 46 | 2 |
| **Cu-H-Beta-300-IE** | 37 | 41 | 2 | 74 | 27 | 2 |

The Cu-modified Cu-H-Beta-25-IE, Cu-H-Beta-150-IE and Cu-H-Beta-300-IE catalysts exhibited a decrease of the Brønsted and Lewis acid sites as compared to the pristine H-Beta-25 catalyst. The plausible explanation for the decrease in the Brønsted and Lewis acid sites in the Cu-modified H-Beta-25, Cu-H-Beta-150 and Cu-H-Beta-300 is the substitution of these sites by CuO (Table 5). The largest decrease in the Brønsted and Lewis acid sites was obtained for the Cu-H-Beta-300-IE catalyst. The lowest amount of tetrahedra Al (IV) present in the H-Beta-300 is the reason for such a low amount of Brønsted and Lewis acid sites. The details of the characterization of the acid sites in H-Beta-25, H-Beta-150, and H-Beta-300 using FTIR-Pyridine and nuclear magnetic resonance (NMR) are given in Ref [40]. Yang et al. proposed a mechanism for the catalytic ozonation of pharmaceuticals including IBU in mesoporous alumina-supported manganese oxide. According to their mechanism, hydroxyl groups are formed via the interaction of water and Lewis acid sites of the catalysts. These hydroxyl groups act as Brønsted acid sites and are able to adsorb ozone on the catalyst surface, on which ozone is transformed to $^\bullet$OH and $^\bullet$O$_3$-catalyst complexes. According to this mechanism, the main active species are hydroxyl radicals [41].

### 2.1.6. X-ray Powder Diffraction (XRD)

X-ray powder diffraction was utilized to study the phase purity and structure of Cu-H-Beta-25-IE, Cu-H-Beta-150-IE, Cu-H-Beta-300-IE, Cu-H-Beta-150-EIM, Cu-H-Beta-150-DP, Ni-H-Beta-25-EIM, Cu-Na-Modernite-12.8-IE, Pd-H-MCM-41-EIM, and Fe-SiO$_2$-DP catalysts (Figure 3).

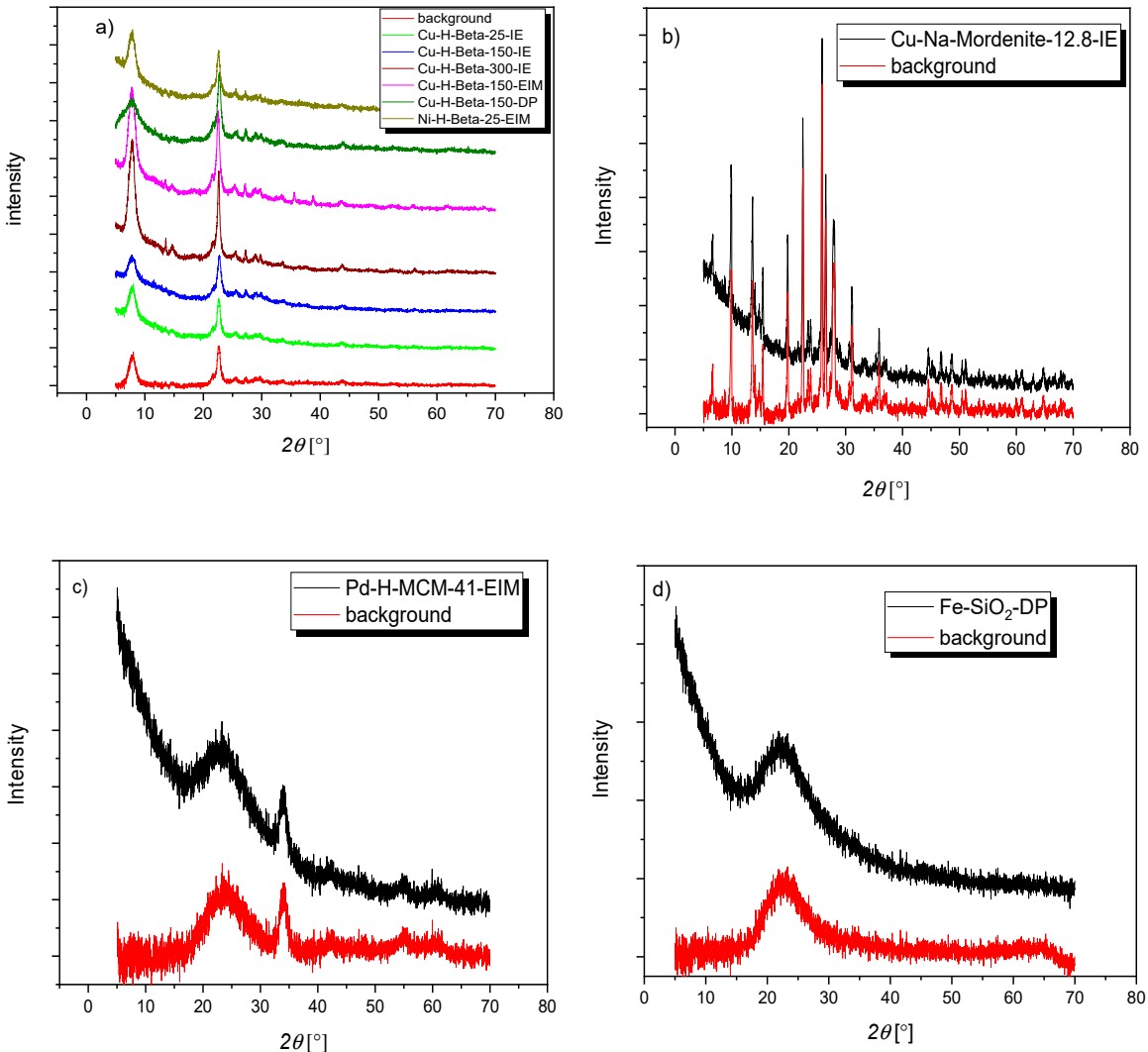

**Figure 3.** X-ray powder diffraction of (**a**) Cu-H-Beta-25-IE, Cu-H-Beta-150-IE, Cu-H-Beta-300-IE, Cu-H-Beta-150-EIM, Cu-H-Beta-150-DP, Ni-H-Beta-25-EIM, (**b**) Cu-Na-Modernite-12.8-IE, (**c**) Pd-H-MCM-41-EIM, and (**d**) Fe-SiO$_2$-DP catalysts.

*2.2. Evaluation of Catalytic Properties in the Degradation of Ibuprofen in Presence of Heterogeneous Catalysts in Combination with Ozonation*

2.2.1. Effect of Different Nitrogen Gas Flow Rate on the Decomposition of IBU

Figure 4 demonstrates the effect of the nitrogen inlet gas flow and temperature on the removal of IBU. Among these experiments, 2.5 mL/min nitrogen shows the highest decomposition rate compared to the experiments carried out without nitrogen and 50 mL/min nitrogen. A slightly higher decomposition rate was observed at 20 °C compared to 5 °C under these conditions. The dissolved ozone concentration at 20 °C, using 450 mL/min and 2.5 mL/min nitrogen was 8.317 mg/L and at 20 °C, using 450 mL/min and 50 mL/min nitrogen was 3 mg/L, which was determined with the indigo method. The ozonator manufacturer proposed to use small amounts of nitrogen in the inlet gas flow to improve the ozonator performance, which indicates that 2.5 mL/min is the optimal flowrate (Figure 4).

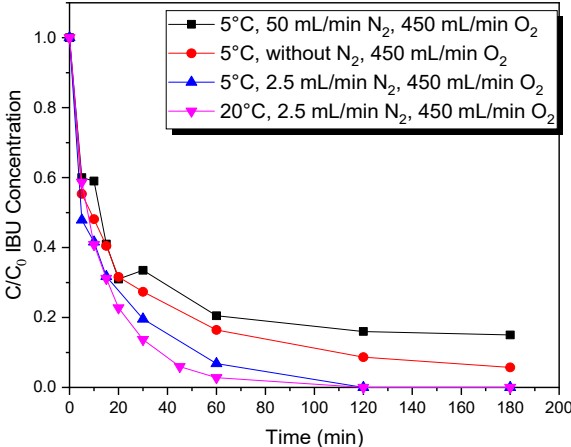

**Figure 4.** The degradation of Ibuprofen [2-(4-isobutyl phenyl) propionic acid] (IBU) by ozonation in absence of catalyst. [IBU] = 10 mg/L, gas flow rate = 450–500 mL/min, T = 20, 5 °C, stirring rate = 1070 rpm.

The removal of IBU and diclofenac in water by ozonation alone, and in combination of photocatalysis with ozonation has been studied by H. Aziz et al. [42], who suggested that ozonation alone contributes to a high energy yield; however, it gives low mineralization of pharmaceuticals. On the other hand, the combination of photocatalysis with ozonation provides a high degradation rate and mineralization for IBU.

The combination of ozonation with heterogeneous catalysis is one of the most practical ways because the catalyst can be simply separated from the solution. This reaction progresses in three possible mechanistic via heterogeneous catalytic ozonation stages: ozone adsorbs on the catalyst surface driving to generate hydroxyl radicals which degrade organic pollutants, organic pollutants adsorb on the catalyst oxidation proceeded via dissolved ozone, and at the end, both ozone and the organic pollutant adsorb on the catalyst with following surface reaction [18]. Additionally, heterogeneous catalysts with sufficient stability and low loss progress the efficiency of the ozonation process. The performance of the catalytic ozonation mainly depends on the type of catalyst, its surface characteristics, and the pH of the solution, which influence the properties of the active sites. Therefore, the crucial step is to select an appropriate catalyst [43].

The degradation of IBU using Cu-modified Beta zeolite catalysts were higher than the non-catalytic one (Figure 5). The explanation for the higher degradation of IBU is attributed to the presence of catalytic active Cu sites, and the presence of Brønsted and Lewis acid sites. Although the largest amount of Brønsted acid sites was determined for H-Beta-25, Cu-H-Beta-25 exhibited a smaller amount of the Brønsted acid sites (Table 5). Hence, it was concluded that it is not only the amount of Brønsted acid sites, but the Cu sites are important for the enhanced degradation of IBU as well. Furthermore, the amount of Cu present in the Beta zeolite, the Cu particle size, and acid sites were associated to the high activity in degradation of IBU. Cu-H-Beta-150-EIM and Cu-H-Beta-150-DP catalysts showed the highest catalytic activity (Figure 5) in the degradation of IBU. In the catalytic ozonation for the removal of IBU proposed by Wang et al. a sludge-Corncob activated carbon was employed as the catalyst. It was reported that the elimination efficiency of IBU in ozone combined with the catalyst is higher compared to the sum of catalyst adsorption and ozonation alone, which is a supportive proof for the catalytic reaction. However, nothing was mentioned about the side products of this treatment [44]. The catalyst activity in the absence of ozone was studied for the degradation of IBU. After two hours, the concentration of IBU did not change at 20 °C, which perhaps indicates that IBU was not adsorbed on the catalyst surface and was not activated in the absence of ozone. Ikhlaq et al. have studied the ozonation of ibuprofen on ZSM-5 zeolites (both of ZSM-5 and H-Beta zeolite catalysts have a high density of acid sites), which revealed the formation of carboxylic acids as by-products, which were not

detected when using ozonation alone [45]. It can be concluded that during the catalytic ozonation in contrast to plain ozonation, an effective oxidative process takes place.

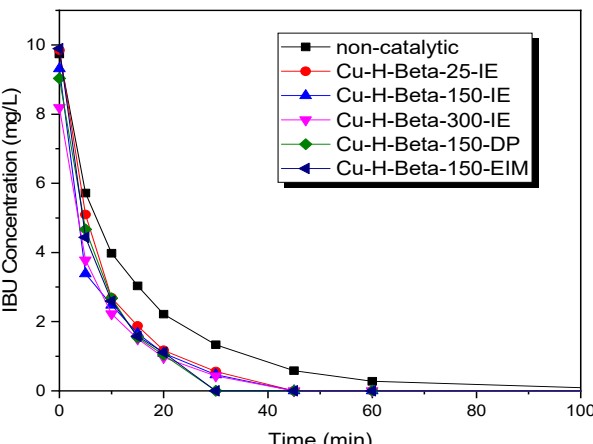

**Figure 5.** The degradation of IBU via catalytic ozonation in the presence of 2.5 mL/min nitrogen. [IBU] = 10 mg/L, gas flow rate = 452.5 mL/min, T = 20 °C, stirring rate= 1070 rpm.

2.2.2. Quantification of Oxidation Products

A quantification method was developed for the intermediate products, which were frequently detected in the ozonation experiments, namely: 1-OH-IBU, 2-OH-IBU, α-OH-IBU, APMP, and 1-OXO-IBU. The main product in all the experiments was 1-OXO-IBU. Between 150 and 330 μg/L of 1-OXO-IBU was formed, which implies that 1–3% of IBU was transformed into 1-OXO-IBU. Similarly, between 0.3% and 0.6% of IBU was transformed into 1-OH-IBU, between 0.2% and 0.5% of IBU was transformed into 2-OH-IBU and up to 0.1% of IBU was transformed into α-OH-IBU. Thus, the total concentrations of the products add up to only a small percentage of the original concentration of IBU. The analysis indicated that a significant amount of IBU is transformed into products which are not detected by LC-MS, such as small organic acids and carbon dioxide. The molecular structure of IBU and the main by-products are displayed in Figure 6.

**Figure 6.** Structural formulas of IBU and the by-products detected.

Figure 7 shows the concentrations of the by-products during the decomposition of IBU. The influence of the catalyst synthesis method on the formation and removal of the by-products can be seen in these figures. The Cu-H-Beta-150-DP catalysts synthesized by the deposition–precipitation technique showed the highest degradation activity of by-products (Figure 7a). The catalytic activity using Cu-H-Beta-150-DP was better in the degradation of 1-OXO-IBU, 1-OH-IBU, 2-OH-IBU, and APMP compared to the other Cu zeolite catalysts and non-catalytic experiments (Figure 7a–d). The highest catalytic activity achieved with Cu-H-Beta-150-DP is attributed to the smallest Cu nanoparticles (4.88 nm, see Table 1). The Cu-H-Beta-150-EIM catalyst revealed an equal activity in the destruction of

1-OXO-IBU and 2-OH-IBU compared to Cu-H-Beta-150-DP, because these two catalysts have similar morphologies (Figure 2d,e). Besides, these results revealed that both the deposition–precipitation and the evaporation impregnation methods were appropriate for the introduction of metallic copper in the catalysts. Cu-H-Beta-25-IE, Cu-H-Beta-150-IE, and Cu-H-Beta-300-IE exhibited a related activity in the destruction of the IBU by-products. Furthermore, Cu-H-Beta-300-IE showed a slightly higher degradation rate compared to the two other catalysts (Cu-H-Beta-25-IE and Cu-H-Beta-150-IE), which was possibly due to the higher surface area and higher crystal size of Cu-H-Beta-300-IE (1013 m$^2$/g).

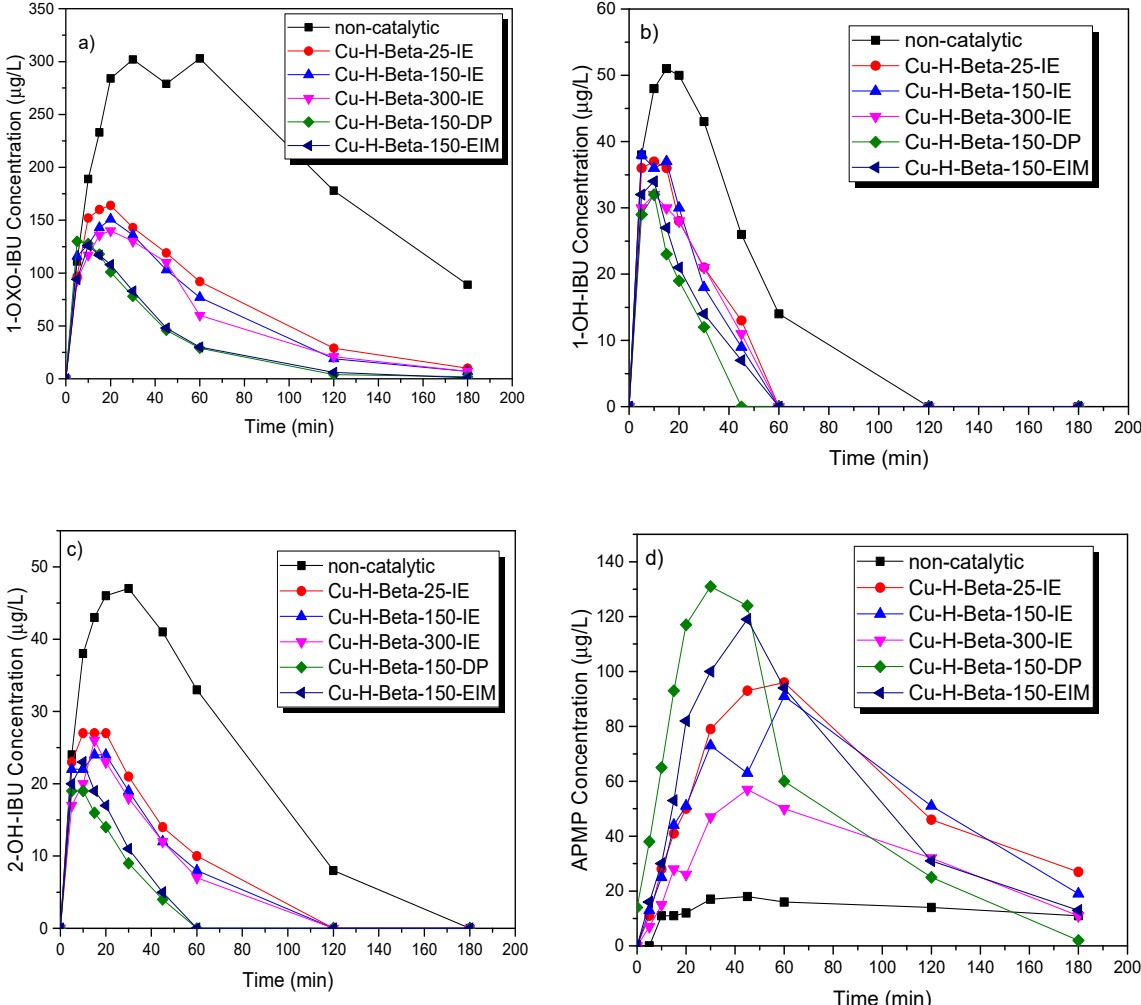

**Figure 7.** (**a**) 1-OXO-IBU, (**b**) 1-OH-IBU, (**c**) 2-OH-IBU, and (**d**) APMP concentration during the decomposition of IBU in the presence of 2.5 mL/min nitrogen. [IBU] = 10 mg/L, gas flow rate = 450 mL/min, T = 20 °C, stirring rate = 1070 rpm.

Figure 8 illustrates the decomposition of IBU in the presence of fresh and regenerated catalysts. As shown by the figure, regenerated catalysts give similar decomposition rates for IBU compared to the fresh catalysts. The regeneration of the Cu-H-Beta-150-DP and Cu-H-Beta-150-EIM spent catalysts were conducted at 400 °C for 120 min. This temperature was sufficient to remove the carboneous deposits (Coke) from Cu-H-Beta-150-DP and Cu-H-Beta-150-EIM. The increase in the surface areas of the regenerated catalysts clearly shows that carboneous deposits (coke) were removed from the catalyst surfaces (Table 3). Temperatures exceeding 400 °C might contribute to sintering Cu nanoparticles, thus deactivating them in the catalytic degradation of ozone. It was observed that Cu-H-Beta-150-DP-regenerated and Cu-H-Beta-150-EIM-regenerated catalysts exhibited a similar catalytic activity in the degradation of ibuprofen to that of the fresh counterparts. The regeneration

and reuse of these catalysts is considerable from the long-term viewpoint of the catalyst stability and cost efficiency.

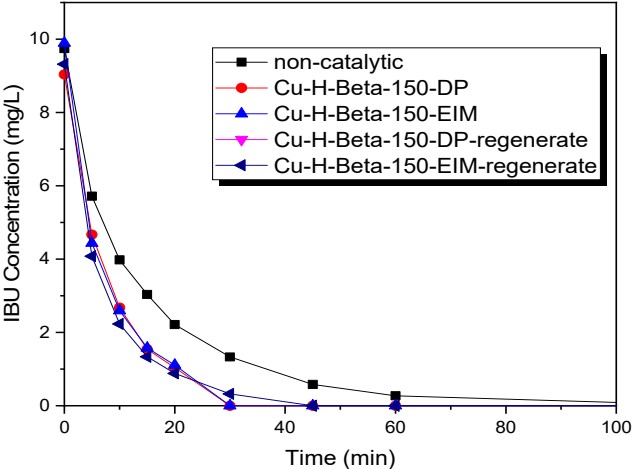

**Figure 8.** The degradation of IBU via catalytic ozonation in the presence of 2.5 mL/min nitrogen. [IBU] = 10 mg/L, gas flow rate = 452.5 mL/min, T = 20 °C, stirring rate = 1070 rpm.

Although the regenerated catalysts exhibited a good stability in the removal of by-products similar to the fresh catalysts (Figure 9a–d) it should be mentioned that the degradation of 1-OXO-IBU (main by-product) was much higher with the Cu-H-Beta-150-DP-fresh, Cu-H-Beta-150-EIM-fresh, Cu-H-Beta-150-DP-regenerated, and Cu-H-Beta-150-EIM-regenrated catalysts than the non-catalytic degradation (Figure 9a–d). These results revealed that the Cu-H-Beta-150-DP and Cu-H-Beta-150-EIM catalysts could be regenerated and reused for the removal of IBU.

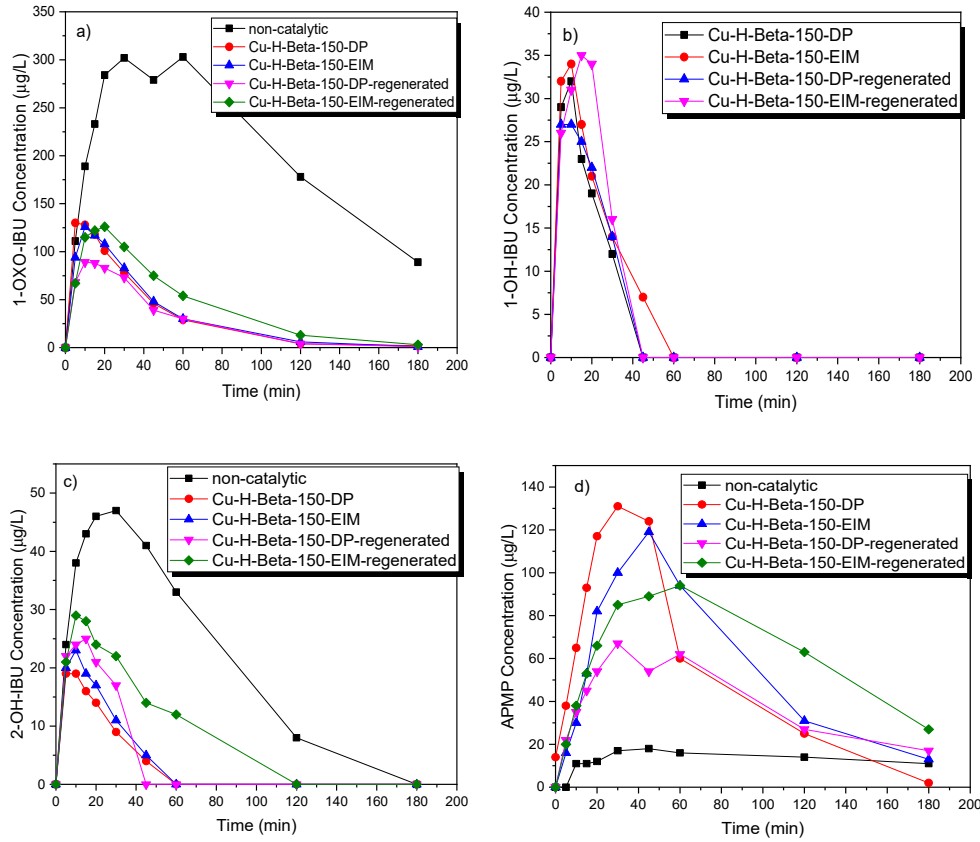

**Figure 9.** (**a**) 1-OXO-IBU, (**b**) 1-OH-IBU, (**c**) 2-OH-IBU, and (**d**) APMP concentration during the decomposition of IBU in the presence of 2.5 mL/min nitrogen. [IBU] = 10 mg/L, gas flow rate = 450 mL/min, T = 20 °C, stirring rate = 1070 rpm.

The degradation of IBU was studied in the presence of several metal-modified catalysts to compare them with Cu-H-Beta-150-DP catalysts (Figure 10). These experiments revealed that the degradation rate of IBU was higher with Cu-H-Beta-150-DP, Cu-Na-Mordenite 12.8-IE, Fe-SiO$_2$-DP, Pd-MCM-41-EIM catalysts than under non-catalytic conditions. Cu-H-Beta-150-DP showed the highest degradation rates of all the catalyst materials studied.

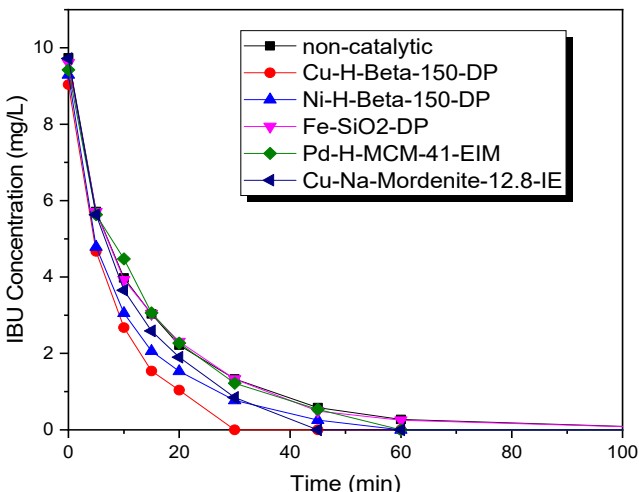

**Figure 10.** The degradation of IBU via catalytic ozonation in the presence of 2.5 mL/min nitrogen. [IBU] = 10 mg/L, gas flow rate = 452.5 mL/min, T = 20 °C, stirring rate = 1070 rpm.

Figure 11 illustrates and compares the effect of different catalysts on the appearance and the degradation of by-products. Cu-H-Beta-150-DP, Cu-Na-Mordenite-12.8-IE, Ni-H-Beta-25-EIM, and Pd-H-MCM-41-EIM showed a higher decomposition rate of 1-OXO-IBU compared to the non-catalytic experiments. Cu-H-Beta-150-DP, Cu-Na-Mordenite-12.8-IE, and Ni-H-Beta-25-EIM showed a higher decomposition rate of 1-OH-IBU compared to the non-catalytic experiments. Cu-H-Beta-150-DP, Cu-Na-Mordenite-12.8-IE, Ni-H-Beta-25-EIM, and Pd-H-MCM-41-EIM showed a higher decomposition rate of 2-OH-IBU compared to the non-catalytic experiment.

These outcomes were in line with the research published by Bing et al., who identified the presence of aliphatic acids—for instance, 2-hydroxy-propanoic acid and glycolic acid from the catalytic ozonation of IBU at the end samples. The results revealed that the catalytic ozonation of IBU proceeds via concurrent hydroxylation; subsequently, the aromatic rings open to form small organic acid molecules toward carbon dioxide and water, wherein the intermediates were generated and degraded at a higher velocity than in the non-catalytic ozonation [46]. These observations confirm that catalytic ozonation has a more effective oxidation achievement compared to ozonation alone for the degradation of the by-products.

The formation of transformation products is dependent on the structure, type of metal, as well as the amount of Brønsted and Lewis acid sites on the solid catalysts (Figure 11a–d). The Cu-H-Beta-150-DP catalyst in the presence of the ozonation reactions for the removal of IBU produced less intermediate transformation products compared to other catalytic processes. This Beta zeolite catalyst with the three dimensional, 12-ring channel and disorder structure, with uniform spherical structured crystals (Figure 2e) exhibited the highest activity compared to the other catalysts screened in this work. On the other hand, copper metal improved the activity of Cu-H-Beta-150-DP compared to Ni-H-Beta-150-DP. Using other catalysts in some cases led to a reduction in the amount of products formed and in some cases to an increase in the amount of products. The amount of products that were formed correlated

with the rate of the IBU transformation so that in the experiments with a more rapid reaction rate, a lower amount of products was created. Cu-Na-Mordenite-12.8-IE additionally exhibited a higher degradation rate compared to other catalysts; this was perhaps due to the copper metal modification and low metal particle size (5.56 nm), which lead to a higher distribution of metal active sites on the support. However, it was not as effective as Cu-H-Beta-150-DP due to the small pore volume (0.158 cm$^3$/g).

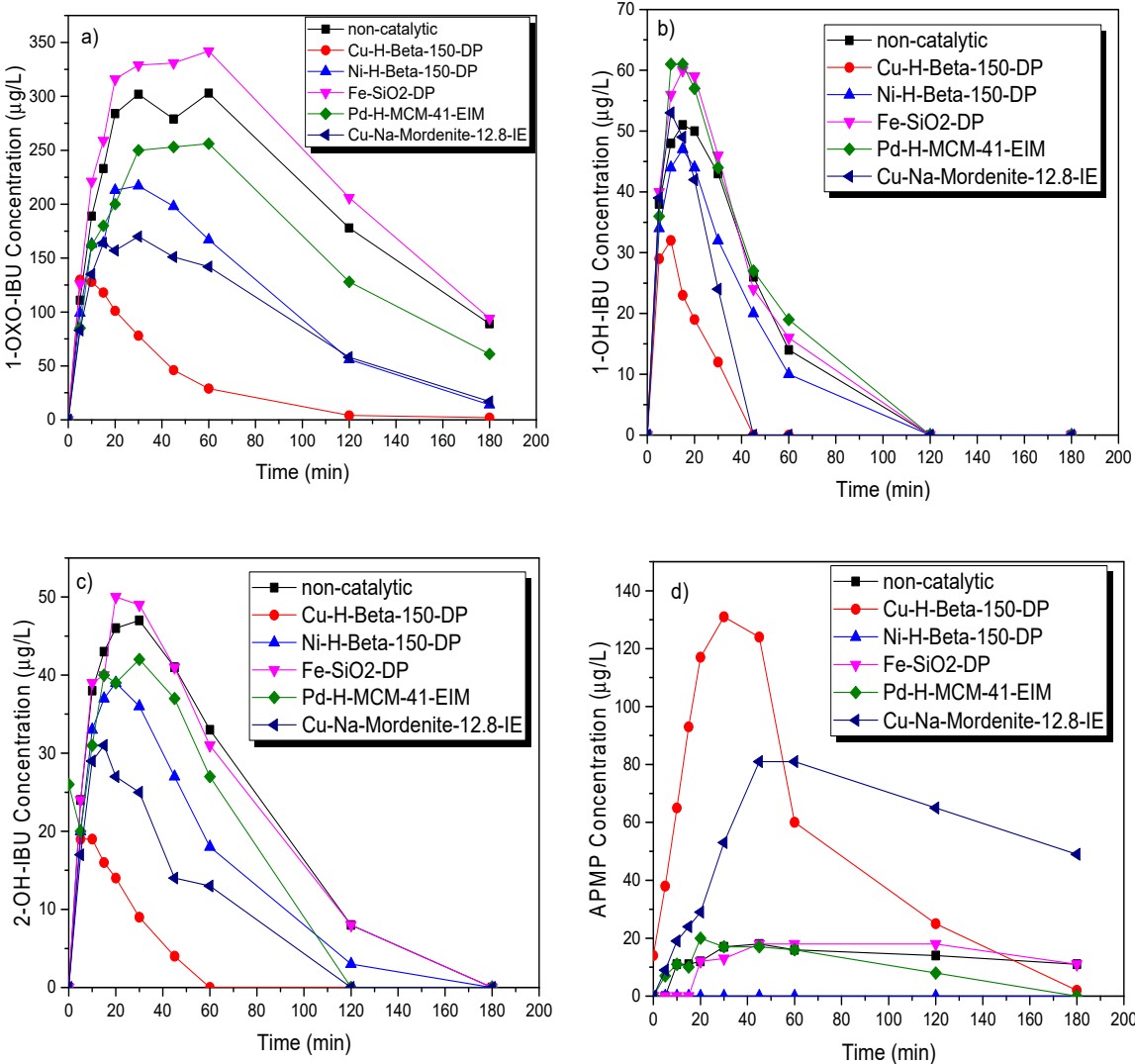

**Figure 11.** (**a**) 1-OXO-IBU, (**b**) 1-OH-IBU, (**c**) 2-OH-IBU and (**d**) APMP concentration during the decomposition of IBU in the presence of 2.5 mL/min nitrogen. [IBU] = 10 mg/L, gas flow rate = 450 mL/min, T = 20 °C, stirring rate = 1070 rpm.

## 3. Materials and Methods

### 3.1. Chemicals

IBU ($C_{13}H_{18}O_2$, MW: 206.28 g/mol, CAS number: 15687-27-1, >98% purity) was purchased from Sigma Life Science (China). HPLC grade methanol ($H_3COH$, MW: 32.04 g/mol, CAS number: 67-56-1) and orto-phosphoric acid 85% ($H_3PO_4$, MW: 98 g/mol, CAS: 7664-38-2) were used. Potassium indigo tri-sulfonate ($C_{16}H_7K_3N_2O_{11}S_3$, MW: 616.72 g/mol, CAS number: 67627-18-3) was provided from Sigma-Aldrich (USA), sodium phosphate monobasic ($H_2NaO_4P$, MW: 119.98 g/mol, CAS number: 7558-80-7) was provided from Sigma life science (Germany). Orto-phosphoric acid was used for the

determination of soluble ozone in the aqueous samples [33]. Ethanol ($C_2H_6O$, MW: 46.06 g/mol, CAS number: 64-17-5, >96% purity) was obtained from Altia (Finland).

## 3.2. Catalyst Preparation

Nine different catalysts—Cu-H-Beta-25-IE, Cu-H-Beta-150-IE, Cu-H-Beta-300-IE, Cu-H-Beta-150-EIM, Cu-H-Beta-150-DP, Cu-Na-Mordenite-12.8-IE, Pd-H-MCM-41-EIM, Fe-SiO$_2$-DP, and Ni-H-Beta-25-EIM—were synthesized. The NH$_4$-Beta-25 zeolite was provided from Zeolyst International. The H-Beta-25 catalyst was obtained using the step calcination procedure of NH$_4$-Beta-25 zeolite. The calcination was carried out in a muffle oven at 450 °C for 240 min. The methods used for the synthesis of metal-modified (Cu-, Fe-, Pd-) catalysts were as follows: evaporation impregnation (EIM), solution ion exchange (IE), and deposition–precipitation (DP). Cu-modified catalysts were prepared, using an aqueous solution of Cu(NO$_3$)$_2$ as a precursor. The Pd-modified MCM-41 catalyst was prepared using aqueous solution of palladium nitrate, whereas aqueous solution of Ferric nitrate was utilized as a precursor for the preparation of the Fe-modified SiO$_2$ catalyst.

The Cu-H-Beta-25-IE catalyst was synthesized by the ion-exchange method, which was carried out in a beaker via an aqueous solution of copper nitrate Cu(NO$_3$)$_2$ and H-Beta-25 at ambient temperature for 24 h. After ion exchange, the catalyst was filtered and washed by two liters of distilled water; then, the Cu-H-Beta-25-IE catalyst was dried at 100 °C. The Cu-H-Beta-25-IE catalyst was calcined at 450 °C in a muffle oven. Cu-H-Beta-150-IE and Cu-H-Beta-300-IE have similar synthesis procedures as aforementioned.

One of the typical catalyst preparation methods for the synthesis of metal modified catalyst is the evaporation impregnation technique employing aqueous solutions of metal nitrate for the preparation of the catalysts. Stekrova et al. used this method for the preparation of H- and Fe-modified zeolite beta catalysts [47]. Cu-H-Beta-150-EIM catalyst was synthesized using the evaporation impregnation technique. The synthesis was carried out using an aqueous solution of Cu(NO$_3$)$_2$ and H-Beta-150 zeolite in a rotavapor. It was rotated for 24 h, during which the aqueous phase was evaporated. The Cu-H-Beta-150-EIM catalyst was dried in an oven overnight at 100 °C. The catalyst was calcined via a muffle oven at 450 °C for 3 h.

One of the most common processes of metal introduction in catalysts is deposition–precipitation. The method is a modification of the precipitation processes in solution. It involves the conversion of a highly soluble metal precursor into a substance with a more limited solubility, which precipitates upon the support [48]. The Cu-H-Beta-150-DP zeolite catalyst was synthesized using the deposition–precipitation technique. The synthesis was carried out in a beaker using Cu(NO$_3$)$_2$ aqueous copper nitrate solution and H-Beta-150. The pH of the aqueous solution was adjusted with NH$_4$OH to 10, at which the synthesis was performed. After 24 h, the Cu–H-Beta-150-DP was filtered and washed via two liters of water, and it was dried in an oven during the night and calcined in a muffle oven at 450 °C for 3 h.

The synthesis of Cu-Na-Mordenite-12.8-IE catalyst was performed by the ion-exchange method utilizing aqueous solution of copper nitrate as a precursor. The synthesis procedure was similar to that of Cu-H-Beta-25-IE, Cu-H-Beta-150-IE, and Cu-H-Beta-300-IE catalysts. The decomposition of the copper nitrate was carried out in a muffle oven at 450 °C for 240 min.

The Pd-H-MCM-41-EIM catalyst was synthesized using the evaporation–impregnation technique. The synthesis was carried out in a flask containing an aqueous solution of Pd(NO$_3$) and H-MCM-41 mesoporous materials. The flask was rotated for 24 h in an evaporator at 60 °C, during which the aqueous phase evaporated and the catalyst was recovered. The catalyst was dried at 100 °C in an oven overnight and calcined via the muffle oven at 400 °C for 180 min.

The Fe-SiO$_2$-DP catalyst was synthesized using the deposition–precipitation technique. The synthesis was carried out in a beaker utilizing an aqueous iron nitrate solution Fe(NO$_3$)$_2$ and SiO$_2$. The pH of the aqueous solution was adjusted with aqueous solution of NH$_4$OH (25%) to pH 10, at which the synthesis was performed. After 24 h, the Fe-SiO$_2$-DP was filtered and washed via two liters

of water. The washing of synthesized catalyst using two liters of distilled water was necessary for the pH neutralization. The catalyst was dried in an oven overnight at 100 °C and calcined in a muffle oven at 450 °C for 180 min.

Ni-modified catalysts were prepared using an aqueous solution of Ni $(NO_3)_2$ as a precursor for nickel. The Ni-H-Beta-25-EIM catalyst was synthesized with the evaporation–impregnation technique. The synthesis was carried out using an aqueous solution of $Ni(NO_3)_2$ and H-Beta-25 zeolite in a rotavapor. It was rotated for 24 h at 60 °C, during which the aqueous phase was evaporated. The Ni-H-Beta-25-EIM catalyst was dried in an oven during the night at 100 °C. The catalyst was calcined by a muffle oven at 450 °C for 180 min.

### 3.3. Physico-Chemical Characterization of Employed Catalyst

The characterization of the catalysts was performed employing transmission electron microscopy (TEM) JEM 1400 Plus, Jeol Ltd, Tokyo, Japan; scanning electron microscopy (SEM) Zeiss Leo Gemini 1530 microscope and energy disperse X-ray analysis (SEM/EDXA), nitrogen physisorption, and Fourier transform infrared spectroscopy (FTIR) ATI Mattson Infinity series, Madison, U.S.A as specified below. The equipment that was utilized to obtain the electron micrographs of the catalysts, metal particle size, and structural properties of catalysts was a JEM 1400 Plus transmission electron microscope by 120 kV accelerating voltage and a resolution of 0.38 nm equipped by OSIS Quemesa 11 Mpix digital camera (rephrase/split) (TEM, model JEM 1400 plus: Jeol Ltd., Tokyo, Japan). The average metal particle size distributions were estimated by counting many particles from the transmission electron graphs. The metal particle size (Cu-, Pd-, Ni-) distribution was given in the form of histograms. The morphology of catalysts was analyzed using SEM (Zeiss Leo Gemini 1530, oberkochen, Germany). The crystallite size of Cu-H-Beta-25-IE, Cu-H-Beta-150-DP, Cu-H-Beta-300-IE, Cu-H-Beta-150-EIM, Cu-Na-Mordenite-12.8-IE, Pd-H-MCM-41-EIM, and Fe-$SiO_2$-DP catalysts were determined using SEM and given in the form of histograms. For the specific surface area and pore volume determination of the catalysts, nitrogen adsorption was employed with the aid of a Carlo Erba Sorptomatic 1900 instrument (Carlo Erba Sorptomatic 1900-Fisons Instruments, Milan, Italy) and calculated with Dubinin and BET equations. Before the measurement, the fresh and regenerated catalysts were outgassed at 150 °C and the spent catalysts were outgassed at 100 °C for 3 h.

The catalyst acidities were estimated with Fourier transform infrared spectroscopy (FTIR, ATI Mattson Infinity series, Madison, U.S.A). The amount of Brønsted and Lewis acid sites were measured by employing pyridine (≥99.5%) as the probe molecule. First, a thin pellet disc of the catalyst was pressed, installed into the FTIR cell, and heated up to 450 °C for 1 h. Then, the temperature was lowered to 100 °C, background spectra of the pellet were recorded, and pyridine was adsorbed on the catalyst sample for 30 min and desorbed consequently by discharge at 250, 350, and 450 °C, correspondingly. The pyridine desorption at 250–350 °C displays weak, medium, and strong sites, 350–450 °C indicate medium and strong sites, and 450 °C indicates strong sites [49]. X-ray diffraction patterns of the catalysts were recorded on a Panalytical X'Pert[3] Powder diffractometer with a $CuK_\alpha$ ($\lambda$ = 1.5406 Å) source. The diffractograms were recorded in the 2θ range of 5–70° in the step size of 0.013° with a count time of 99 s at each step.

### 3.4. Experiment Method for Ozonation Activity and Kinetics

The kinetic experiments were conducted in a double jacket glass reactor operated in semi-batch mode, connected to an ozone generator. It is advantageous to utilize the ozone generator to provide ozone, because an ozone generator can produce functionally and stable ozone in situ [19]. Via a 7 μm disperser at the lowest point of the reactor, an ozone gas mixture was constantly bubbled into the mixed liquor including IBU, ethanol (used as a diluting solvent because of the low solubility of IBU in pure water), and deionized water, which affords the semi-batch mode reactor. To provide vigorous mixing of the liquid phase, a Spinchem$^{TM}$ rotating bed stirrer was used. The ozonation process was carried out with 1000 mL of solution, 10 mg/L of IBU concentration, 10 mL/L ethanol, 450–500 mL/min

gas flow, 1070 rpm mixing rate, 5–20 °C reactor temperature, and 3 h reaction time. Using a high concentration of IBU (10 mg/L) was due to enabling the identification of transformed intermediates and by-products at low concentration during experiments. The ozonator manufacturer requires the use of a small amount of $N_2$ (0.5–5%) in the feed for the high-grade performance of the ozone generator. When an oxygen gas flow rate of 0.450 L/min combined with 0.05, 0.0025, and 0 L/min $N_2$ (super-dry feed gas dew point—60 °C), the ozone generator (Absolute Ozone, Nano model, Edmonton, AB, Canada) generated around 60 mg/L concentration of ozone in the gas phase. For pH measurement, the pH-stat device (tiamo$^{TM}$, Metrohm, Herisau, Switzerland) was used, and the pH of the solution was approximately 5, but after 15 min, the ozonation pH dropped to 4.5 and later slowly decreased to 3.3 during the experiments. Then, 0.5 g of catalysts were immobilized within the rotating bed stirrer and the catalyst particle sizes were between 150 and 500 μm; these particles remained inside the stirrer pretty well. Samples were withdrawn before, during, and at the end of the experiments [35]. A general schematic view of the experimental apparatus is presented in Figure 12.

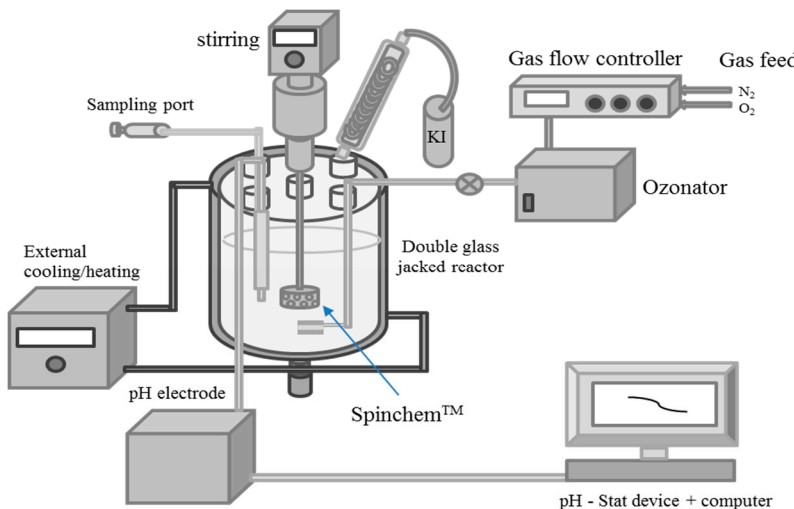

**Figure 12.** Scheme of the semi-batch reactor system for the evaluation of heterogeneous catalysts in the degradation of ibuprofen.

*3.5. Chemical Analysis*

IBU was determined via an HPLC (Agilent Technologies 1100 series) with a UV-Vis photo diode array detector set at 214 nm, and a quaternary pump. The column used was an Ultra Techsphere ODS-5u-(C18), 250 mm × 4.6 mm. The mobile phase consisting of a 70:30 mixture of methanol and 0.5% phosphoric acid (pH: 1.8) was flowing at 1 mL/min; the sample injection volume and retention time were 20 μL and 10 min, sequentially [35].

For the LC-MS/MS spectrometry, an Agilent 6460 triple quadrupole mass spectrometer equipped with an Agilent Jet Spray electrospray ionization (ESI) source was employed in multiple reaction monitoring (MRM) mode. Nitrogen was used as drying gas, sheath gas, nebulizer gas, and collision gas. Drying gas and sheath gas were kept at 11 and 12 L/min, respectively and heated to 350 °C. The nebulizer pressure was set to 25 psi. A capillary voltage of 4500 V and a nozzle voltage of 1500 V were utilized. The compounds were analyzed in positive and negative ionization modes. The fragmentor voltage and collision energy were optimized for both compounds individually using the MassHunter Optimizer software (Table 6). An accelerator voltage of 3 V was used. The chromatographic separation was made using an Agilent 1290 binary pump equipped with a vacuum degasser, an autosampler, a thermostatted column oven set to 30 °C, and a Waters xbrigde C18 column (2.1 × 50 mm, 3 μm). The eluents were 0.1% formic acid in water (A) and 0.1% formic acid in acetonitrile (B). Initially, the composition was held at 5% (B) for 0.5 min; then, the composition was increased linearly to 95% (B) over 3 min. The eluent composition was held at 95% (B) for 0.5 min before being returned to the primary conditions over the

next 0.1 min and given 1.4 min for equilibration. The flow rate was 0.4 mL/min. The injection volume was 10 μL. The internal standard system was used for quantification.

**Table 6.** Mass spectrometer parameters for the ibuprofen oxidation products.

| Compound | Precursor Ion | Product Ion | Fragmentor (V) | Collision Energy (V) | Polarity |
|---|---|---|---|---|---|
| 1-oxo-Ibuprofen | 221.1 | 175.1 | 70 | 8 | positive |
| | | 133.1 | 70 | 16 | positive |
| 1-OH-Ibuprofen | 221.3 | 159.1 | 70 | 4 | negative |
| | | 143 | 70 | 16 | negative |
| 2-OH-Ibuprofen | 221.3 | 177.1 | 65 | 0 | negative |
| | | 159.1 | 65 | 4 | negative |
| α-OH-Ibuprofen | 221.1 | 175.1 | 55 | 16 | negative |
| | | 133.1 | 55 | 20 | negative |
| 2-OH-Ibuprofen-d6 | 227.3 | 183.2 | 65 | 0 | negative |

## 4. Conclusions

Cu-modified Cu-H-Beta-25-IE, Cu-H-Beta-150-IE, Cu-H-Beta-300-IE, Cu-Na-Mordenite, Ni-H-Beta-25-EIM, Fe-modified $SiO_2$-DP, and Pd-modified H-MCM-41 catalysts were successfully synthesized and used for the degradation of ibuprofen in presence of ozone as the oxidizing agent. The method of introduction of Cu- in H-Beta zeolite, the particle size of Cu, and acid sites were observed to influence the degradation of ibuprofen. The DP and EIM preparation methods exhibited a higher performance catalysts compared to the IE preparation method. Furthermore, the type and structure of the support materials used for the catalyst synthesis of beta, mordenite zeolites, MCM-41 mesoporous material, and $SiO_2$ were of an immense importance for the catalytic activity in the degradation of ibuprofen from this comparison; the H-Beta structure revealed the highest activity, while $SiO_2$ exhibited the lowest activity.

The results revealed that IBU was successfully decomposed by ozone in the absence of the optimal catalyst operated at 20 °C, 450 mL/min oxygen, and 2.5 mL/min nitrogen within one hour. In catalytic experiments, Cu-H-Beta-150-EIM and Cu-H-Beta-150-DP showed the highest degradation rates, and IBU degraded entirely within 30 min. Liquid chromatography-mass spectrometry was used to quantify by-products at very low concentration levels. Most of the catalysts were useful in the elimination of the by-products. Cu-modified catalysts were the most effective in the removal of the by-products, especially Cu-H-Beta-150-DP. This catalyst exhibited a low average particle size for Cu around 4.88 nm with a high pore volume (0.259 $cm^3$/g). Moreover, the H-Beta zeolite is a hydrophobic catalyst that can attach organic IBU from water and steer the heterogeneous catalysis in the presence of ozone, which makes Cu-H-Beat-150-DP a suitable catalyst for the destruction of IBU. Thus, it can be concluded that the role of copper modification on the catalyst is important in the degradation of IBU. Nevertheless, the metal particle size of copper, its dispersion, and the amount of copper are also taking part in the degradation reaction. Hence, one has to take into consideration all the above significant facts while elaborating on the explanations for IBU degradation. Furthermore, the regenerated Cu-H-Beta-150-DP catalysts were also useable for the degradation of IBU.

**Author Contributions:** S.S. catalytic ozonation experiments, catalyst preparation, characterization and writing, M.K. by-product quantification, L.M.: experimental, senior scientists and supervisors with the following competences: P.T.: ozonation technology, N.K.: catalyst specialist, K.E.: reactor design, M.P.: TEM expert, J.-P.M.: stirrer expert, L.K.: analysis of organic components in aqueous environment, P.E.: organic reaction technology. T.S.: chemical kinetics and experimental planning. All authors have read and agreed to the published version of the manuscript.

**Funding:** This research received no external funding.

**Acknowledgments:** This study is a part of the research of the Johan Gadolin Process Chemistry Centre, which is a centre of excellence supported via Åbo Akademi. Financial support from Svenska Litteratursällskapet (SLS), Centre for International Mobility (CIMO) and Tekniikan Edistämissäätiö (TES) is appreciatively acknowledged. SpinChemTM AB is appreciatively acknowledged for providing the RBR equipment. The Bio4Energy programme

in Sweden is acknowledged. Walter och Lisi Wahls Stiftelse för naturvetenskaplig forskning is acknowledged for funding the purchase of the ozonator generator.

**Conflicts of Interest:** The authors declare no conflict of interest.

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
