# Peer review of "Synthesis and Characterization of Metal Modified Catalysts for Decomposition of Ibuprofen from Aqueous Solutions"

_catalysts, doi:10.3390/catal10070786_

Round 1
Reviewer 1 Report
The reviewed work is a kind of summary in which the effectiveness of various catalysts in removing ibuprofen from the aquatic environment is examined. My comments on the work relate primarily to methodological issues.
- Table 2 and Table 3 should be presented under chapter 2.1.2 and not under 2.1.3.
- Lines 203-204. "And the 203 largest metal content was obtained for Ni-H-Beta-25-EIM (Table 3)." There should be "(Table 2)". Table 3 does not show the results of chemical contaminants analysis
- I consider using the EDS method to analyze metal content as a serious methodological error. EDS is completely unsuitable for quantitative analysis of the chemical composition of fine dispersion materials. The results of the analyzes may be subject to huge mistakes. I recommend using the WDS or micro-XRF or ICP methods
- Why was the XRD method not used to analyze the phase composition of the catalysts? XRD is the primary method for documenting the phase composition diversity of compared materials - in this case, catalysts. In fact, without using the XRD method, the authors are unable to demonstrate that they obtained such catalysts that they assumed.
- Descriptions of catalyst synthesis methods are not precise. There is no information on the concentration of reagents used (metal nitrates, ammonia). Information on washing the synthesis products with 2 liters of redistilled water appears. Why exactly 2 liters? What was the measure of washing efficiency?
- Keywords. The first four words are a repetition of words from the title of work. Please replace them with others.
Author Response
Please find reply to the comments of reviewer 1

Reviewer 2 Report
All comments are in attached file.

Author Response
Please find reply to the comments of reviewer 2

Round 2
Reviewer 2 Report
The authors did not give a satisfactory answer to any serious comments. The new version of the manuscript contains almost no changes. The reviewer still considers the comments about methodology of the study to be serious and relevant; accordingly, it makes no sense to review it again.
Author Response
Please find reply to reviewer 2

Round 3
Reviewer 2 Report
Almost nothing was changed.
Author Response
The comments of the reviewer 3 regarding english language and spellings have been taken into consideration in the revised version of the manuscript.
